# Efficacy of a Contextualized Measurement of Life Satisfaction: A Pilot Study on the Assessment of Progress in Eating Disorder Therapy

**DOI:** 10.3390/ijerph192114452

**Published:** 2022-11-04

**Authors:** Maria Aymerich, Antoni Castelló, Ramon Cladellas

**Affiliations:** 1Institute of Research on Quality of Life, University of Girona, 17004 Girona, Spain; 2Department of Basic, Developmental and Educational Psychology, Autonomous University of Barcelona, 08193 Barcelona, Spain

**Keywords:** eating disorders, validation, assessment, life satisfaction, adolescent, women

## Abstract

Eating disorders strongly affect psychological distress and its perception. However, most of the existing instruments for assessing life satisfaction rely on a point-estimation method that is biased due to the circumstantial conditions around the time of assessment. The main goal of this study was to apply a different kind of instrument—the Life Satisfaction Chart—that situates the current state of life satisfaction in the context of personal history and describes the life stages through a graph. The assessment was applied to a sample of 29 adolescent women (average age of 17.88) who were enrolled in a clinical program to treat their eating disorders. The results showed that their estimation of their current life satisfaction was almost identical to the estimation provided by a therapist for those who were in therapy phases 1, 2, and 3 (of four), while patients’ point-estimation satisfaction showed statistically significant differences when compared with the situated estimations. In therapy phase 4, significant discrepancies were observed between the therapist’s perception and the patients’ perception, because the therapist focused only on eating disorder recovery, whilst the patients evaluated their lives under almost-normal conditions, taking into account further dimensions. The Life Satisfaction Chart is a new approach to life-satisfaction measurement that showed promising measurement and therapeutical properties.

## 1. Introduction

Young women are increasingly confronted with various problems that occur mainly during adolescence and young adulthood. This has led to an increase in mental problems at these ages, to the point that a global public health crisis concerning adolescents’ mental health has been reported [1,2]. Such is the case of eating disorders (EDs), which are characterized by behavioral and cognitive aspects that result in the significant impairment of an individual’s wellbeing [3].

Proper diagnosis and treatment are decisive in improving the prognosis of these patients [4]. In this process of assessment and psychotherapeutic intervention, the way in which a patient perceives or reconstructs the course of her life is highly relevant information.

A good assessment of the problem enables the implementation of an intervention by a therapeutic team and is associated with a better prognosis after treatment, while poor detection can lead to the chronicity of the disorder [5].

This paper aimed to assess the importance and validity, in the context of the clinical treatment of patients with eating disorders, of a POLS (Present Overall Life Satisfaction) indicator, namely the Life Satisfaction Chart (LSCh), which includes the perspective of the personal-history time axis obtained by means of a novel graphical technique. The LSCh was developed by the integration of two previous techniques, the “Life Line” [6,7,8,9,10,11,12,13,14] and the “Life Chart Methodology-retrospective (LShM-r)” [15,16,17,18,19,20,21,22], after adapting them to the measurements of a client or patient’s degree of wellbeing and suffering.

As a result, the LSCh takes the form of a graphic representation, retrospective in nature, of the flow and evolution of a patient’s levels of life satisfaction and dissatisfaction throughout their life, or throughout a specific period of treatment, relating them to the conditions and events the patient has experienced, according to their subjective perception. An example of this can be seen in Figure 1.

This technique is of great interest for the patient, the therapist, and the therapeutic process itself. For the patient, the LSCh technique is intuitive, enjoyable, and simple. The chronological review of events experienced and the graphic representation of the impact, positive or negative, that they have had on personal wellbeing facilitates the pursuit of introspection and individual reflection by the patient in the therapeutic environment. This exercise of representing the LSCh prepares the patient for the therapeutic sessions by focusing on the analysis of the events that have been significant to their wellbeing throughout their life history. This helps to avoid the risk that either the therapist or the patient themself will focus on irrelevant aspects of the patient’s life history. According to Harding [23], the self and one’s life history are in part created and enacted through the interview, and the interviewer has a lot of power over this process. The LSCh realization exercise directs the patient to visibly represent their own idiosyncratic trajectory, which allows both patient and therapist to share the same complete and self-validated representation of the case to be treated.

The usefulness of the technique for the therapist is important, whether the LSCh technique is used during the anamnesis or whether it is also used to monitor the patient’s evolution during treatment. As a complement to the anamnesis, the LSCh allows the therapist to gain a better understanding of the patient’s life history, ascertaining what events they have experienced, linking these events to the impact that they have had on the patient, and determining whether the patient has overcome these impacts and how long it took them to recover. It also allows the therapist to learn what positive events have helped the patient feel good and satisfied, opening the door to recovery by using these events during the therapeutic process. This guides and directs the therapist to explore certain parts of the patient’s life history in a meaningful way [24].

Finally, the LSCh shows how the patient feels in the present, i.e., the moment at which they are seeking therapy, by offering a POLS (Present Overall Life Satisfaction) value [25]. The reuse of the technique at some point during the therapy or at the time of its completion can provide information to the therapist that enables the patient’s progress and recovery of wellbeing to be monitored. The recovery of levels of emotional well-being and life satisfaction at the end of therapy are indicators of the effectiveness of the therapy in the treatment of multiple issues and disorders, such as depression.

The advantages of using the LSCh at the beginning of therapy are that it promotes greater patient–therapist interaction by allowing both parties to discuss the information represented graphically by the patient and to expand on and clarify any of the points included. These are all opportunities to develop a good rapport between practitioner and patient [8,26] and prepare both for further therapeutic work, as well as to establish a shared trajectory.

Moreover, the observation of the graph ultimately obtained by means of the LSCh is also in itself very revealing for the patient, as several subjects have stated after the administration of the tool. In fact, this author of [27] has already underlined the importance of life diagrams, in that we live in a visual world and, currently, no topic, field of study, or discipline is immune to the influence of researchers (and clinics) adopting visual perspectives.

The combined use of visual methods with qualitative research is becoming more and more frequent, as it presents several advantages [28]. Firstly, the use of visual methods (e.g., photographs, drawings, or other visual resources) can overcome some communication barriers, such as in the treatment of autism [29]. Furthermore, the representation of a graph, as a visual element, presents a wide range of benefits compared to verbal representation in the diagnosis and treatment of mental-health-related problems. Graphic representation favors the development of feelings, encourages willingness to be guided by one’s feelings, supports the achievement of challenges and accomplishments, and changes the meaning of experiences [30]. In this way, graphic representation brings out dimensions that are not easily represented by language. Ratios or differences in intensity are much more accurately expressed graphically. For example, if a person declares that at a certain stage in their life “their wellbeing decreased”, this statement could correspond to multiple graphic representations, such as (on a scale of 0 to 10) a transition from 9 to 7, from 5 to 3, or from 8 to 2. Therefore, although these three examples (among many other possible representations) entail a decrease in wellbeing, their visualization reveals information that goes beyond the decrease itself: the intensity of the decrease or the level of wellbeing in which it results. Thus, the first example indicates a moderate decline, although the wellbeing remains in the high range; the second example indicates an equivalent decline, but in a lower range; and the last example indicates a very intense decline that involves a shift from the high to the low range. While individual changes could be well expressed numerically, the sequence of changes across different life stages is much better represented graphically, bringing a holistic perspective that considers trends, length of time, overall stability, and so on.

In the same vein, given its status as a visual technique, a strength of the LSCh is its ability to provide distance between the experiences of intimate events and their narration [31]. Overcoming the limits of verbal language allows for the in-depth reflection on issues that might otherwise remain incomplete or hidden [32,33].

The LSCh has demonstrated its usefulness and benefits in studies on wellbeing and life satisfaction [25,34,35] and life history research in the field of social sciences [24]. In their article, Aymerich et al. [34] administered the LSCh to study gender differences in the perception of how life satisfaction evolves throughout childhood and adolescence. The data indicated that life satisfaction reported retrospectively by young people showed a significant decrease from the age of 11 onwards, and this was much more noticeable in the cohort of girls. In another study, Aymerich et al. [25] confirmed that Present Overall Life Satisfaction (POLS) is a suitable indicator for measuring subjective wellbeing in adolescents, since it had good convergent validity with other instruments of subjective wellbeing and good divergent validity with anxiety and (especially) depression. Furthermore, it is also worth mentioning that the POLS indicator showed moderate correlation with context-free SWB multi-thematic verbal scales, which indicated that without incorporating a prior reflection on one’s personal biographical history by means of the LSCh, the exact same aspects are not evaluated. Another study [35] administered the LSCh to recently retired people to study the process of psychological adaptation to retirement. The technique allowed the researchers to distinguish different types of adaptive profiles in terms of increased or decreased levels of satisfaction experienced throughout the process of adjusting to retirement, as well as the time required to properly adapt to the new retired status, differentiating between simple and brief adaptations and those that were painful and complicated. Additionally, research in the social sciences has presented life diagrams as a useful methodological and analytical tool, as they have helped researchers and research participants communicate better and more extensively about the life story in question, while still empowering the research participants [24]. Although the LSCh generally appears to be a tool of great interest and applicability for clinical settings, there are no studies to date that have guaranteed its validity and reliability in this particular context.

Despite the theoretical potential and general practical advantages of the instrument, it needs to be empirically validated in the psychotherapeutic setting. To this end, based on a sample of patients in treatment for eating disorders, the present work focused on the following objectives.

### Research Goals

(1)To establish the validity of the LSCh, through the POLS indicator, contrasting it with the therapist’s assessments (external validity) and with the patients’ own estimates of current satisfaction (concurrent validity).(2)To demonstrate that the POLS measure obtained from the LSCh is more accurate and reliable than decontextualized point-in-time measurements of life satisfaction.

## 2. Materials and Methods

### 2.1. Participants

The sample selection was purposive and included all the patients with eating disorders from a single center, the Unitat de Patologies Alimentàries of the Clínica Bofill (UPA; corresponding to an eating disorders unit (EDU)). This is a specialized health center for the diagnosis and treatment of patients affected by eating disorders that works on an in-patient day-hospital basis.

A total of 29 young females were diagnosed with an ED, aged between 16 and 22 (M = 17.88, SD = 2.64). Most of them were presenting their first episode of eating disorders (n = 27), and the remaining two were presenting their second episode. Prior to their admission to the eating disorders unit, 28 of them had tried out-patient treatment at the same center, with a mean duration of 6.79 months (SD = 7.95 months). Since they did not achieve an improvement in their disorder, they were admitted to the in-patient treatment. The time spent in the EDU when data were collected ranged from 15 days to 54 months (M = 20.38 months, SD = 15.25 months).

At the time when they were admitted as in-patients, participants met DSM-5 [36] criteria. More specifically, twenty-one met the criteria for anorexia nervosa restrictive subtype, five met the criteria for the binge/purge subtype, two for bulimia nervosa, and one for binge-eating disorder.

Table 1 shows the distribution of patients by the level of evolution/treatment phase, where 1 is the initial level of treatment on admission and 4 is the level of follow-up, post-hospitalization, and pre-discharge.

### 2.2. Instruments

#### 2.2.1. Overall Life Satisfaction [37]

The Overall Life Satisfaction (OLS) scale is a context-free scale used in life satisfaction research [38,39] that assesses life satisfaction or subjective wellbeing using a single item. The patient is asked: *“On a scale of “0 to 10”, where “0” is “very dissatisfied with my life” and “10” is “very satisfied with my life”, where would you currently place yourself?”*

Each of the patients answered the OLS scale question. Likewise, the therapist also answered the OLS scale question for each of the patients, according to their professional knowledge.

#### 2.2.2. Present Overall Life Satisfaction (POLS) [23] Based on the LSCh for Patients

The LSCh is a technique for evaluating subjective wellbeing and its informed evolution; it involves the subject considering the passage of time and events that have taken place in their life (see Figure 1). POLS, a single-item psychometric scale that is determined by considering the present-day value of the subject’s score on the Life Satisfaction Chart (LSCh) [25], was considered within this scale.

To obtain the POLS, the patient was presented with the LSCh representation sheet and was informed that we wanted her to carry out an exercise to graphically represent how she considered her life satisfaction and wellbeing had evolved, starting from her earliest memories of how she felt.

The LSCh profile sheet was framed by two coordinates. On the vertical axis, the possible levels of satisfaction were represented on a scale from 0 to 10, with values from 0 to 5 representing levels of dissatisfaction/discontent (colored in orange) and values from 5 to 10 representing levels of satisfaction/wellbeing (colored in blue). On the horizontal axis, the patient’s age was represented, which by default ranged from 6 to 20; however, we explained to the patient that she did not necessarily have to start at age 6, but rather from whenever she clearly remembered how she felt about her life, up to the present moment. She was informed that there were no right or wrong answers and that her representation would have no impact on her treatment, as the information was confidential. At the same time, she was given a pencil and an eraser so that she could erase and rectify what she had written down as many times as she considered necessary until she had represented how she considered her levels of wellbeing or discomfort had evolved throughout her life.

The patient was allowed to carry out the exercise without intervention from the therapist, unless this was requested to clarify any doubts.

Once the patient had finished her representation, the therapist asked the patient to write down the events associated with the different stages of wellbeing/discomfort, with the possibility of rectifying the representation of the LSCh if she deemed it necessary while recalling these reasons, with the aim of making the final representation of her LSCh more accurate.

At the end, she was thanked for her participation and cooperation in the study and was offered the opportunity to provide her opinion on the exercise and suggestions for possible improvements based on her experience.

### 2.3. Variables

The variables that were analyzed to achieve the proposed objectives were as follows:

Internal variables (self-assessed by the patient)

OLS-Patient: current life satisfaction level provided by the patient through the OLS scale.

Patients who were affected by the disorder and were aware of their psychological and emotional situation assessed their current degree of emotional wellbeing or discomfort using the single-item, decontextualized OLS scale.

POLS: current life satisfaction level provided by the patient through POLS based on the LSCh.

The patients, after representing in a contextualized manner how their life satisfaction had evolved throughout their childhood and adolescence using the LSCh, finished the graph by contextualizing their degree of life satisfaction at present, or their POLS value.

External variables (assessed by therapist)

Patient’s level of treatment

The patients diagnosed and undergoing treatment for an ED in the study were assigned by the therapist to 4 possible levels of treatment, according to the degree of severity and evolution of their condition under treatment.

Level 1 patients were newly admitted patients, who denied their illness, resisted treatment, and needed constant and extensive symptomatologic control.Level 2 patients were patients with an awareness of their illness, who accepted the professional symptomatologic control and cooperated with treatment. They were granted greater freedom to leave the center on some afternoons.Level 3 patients presented good self-management of symptoms; improved in group and individual psychotherapy; collaborated as informal cotherapists with Level 1 and 2 patients; and had extended permission to leave the center, attending therapy sessions only in the mornings or in the evenings.Patients in 4 were recovered patients in the pre-discharge phase, who already led an autonomous life outside the center and only attended once a month for follow-up sessions before full discharge.

Time since admission was not a variable to be considered, given that improvement in treatment does not depend directly and quantitatively on the time a patient spends in treatment, but rather on many other variables that represent the complexity of the disorder (the presence of comorbid psychiatric disorders, the presence of stressful situations, the level of social and family support in accordance with hospital treatment guidelines, etc.). Therefore, for this research, the level of treatment assigned by the clinicians was considered as an indicator of the therapeutic improvement of the patients.

OLS-Therapist: patient’s current life satisfaction according to the therapist

Patients were also evaluated by the therapist in charge of the psychological intervention unit, who coordinated and supervised the team of four psychotherapists and one psychiatrist who treated the patients. As a professional with in-depth knowledge of the disorder and of the psychological and emotional situation of each patient in treatment at the center, the head therapist assessed the degree of emotional wellbeing or discomfort of each patient at the time of admission and at the time of the study.

### 2.4. Procedure

The management of the health center were contacted to inform those in charge of the study’s interest in their collaboration. Once consent was obtained from the center’s management, they informed the potential participants of the study and their families and obtained final consent for participation. All the patients contacted agreed to take part in the study. The overall research project to which this study belongs was approved by the Ethical Committee on Human and Animal Experimentation at the Autonomous University of Barcelona. The general research project was CEEAH 3850 and was entitled “Analysis of the links between negative affect, emotional regulation, and risk of suicide in youngsters”.

The administration of the instruments was carried out in the same care center on an individual basis. At the beginning of the session, participants were again informed of the objective of the study. It was stressed that there was no link to the therapeutic process in the center or its professionals, and participants were guaranteed anonymity and the confidentiality of information. Each patient was allotted the necessary time for data collection, which ranged from 15 to 25 min.

As one of the measures for checking the validity of the information provided by participants, patients were also evaluated by the therapist responsible for the psychological intervention unit, who coordinated and directed the team of psychotherapists and psychiatrists who treated the patients. The head therapist assessed the level of life satisfaction they considered each patient had been experiencing at the time of admission to the center and then at the time of the study. There was no contact between the information collected from the patients and that provided by the therapist.

From this information, only the assessment made by the therapist at the time of the study was taken into consideration, as the level of satisfaction attributed to all patients at the time of admission was the same (the minimum possible value that could be given, 1). This did not provide any complementary information that could help in the evaluation process of the tool used.

### 2.5. Statistical Analysis Overview

Analyses were conducted using non-parametric tests due to the number of participants in the sample.

In a first analysis, a non-parametric test of k independent samples (Kruskal–Wallis) was performed considering the level-of-treatment variable (1 to 4) as the grouping variable and therapist-reported patient satisfaction and patient-perceived satisfaction as dependent variables.

A comparison between two levels to determine possible significance was carried out using the non-parametric Mann–Whitney test. In addition, for each of the stages, the Wilcoxon test was applied with paired data, contrasting the two assessments of the patients (OLS and POLS).

Next, several analyses related to the Wilcoxon non-parametric test were carried out between the variables—therapist-reported patient satisfaction, OLS patient-perceived satisfaction, and POLS patient-perceived satisfaction.

Finally, Spearman’s correlations between the three variables were computed.

The following statistical tests were carried out to determine whether the objectives of this work were met.

## 3. Results

### 3.1. Objective 1

The first objective was to analyze whether the perception of the therapist (OLS) and of the patient (OLS and POLS) at the time the research was carried out varied according to the level or phase of treatment the patients were in, as it is shown in Table 2.

The OLS-Therapist variable was used primarily to demonstrate how the different scores provided by the therapist served to place the patients into different stages or levels. Thus, the results were congruent in that the life satisfaction score increased as the stage or phase progressed.

The results obtained by means of the contrast tests were as follows:

For OLS-Therapist, significant differences were observed between Level 1 and the other levels (between Levels 1 and 2, *p* = 0.010; between Levels 1 and 3, *p* = 0.002; between Levels 1 and 4, *p* < 0.001). Significant differences were also observed between Level 2 and Level 4 (*p* = 0.009).

For OLS-Patient, significant differences were observed between Level 1 and Levels 2 (*p* = 0.007) and 3 (*p* = 0.007).

For POLS (patient only), significant differences were observed between Level 1 and the other levels (between Levels 1 and 2, *p* = 0.007; between Levels 1 and 3, *p* = 0.003; and between Levels 1 and 4, *p* = 0.013).

Figure 2 shows that the therapist’s and patients’ ratings coincided with the POLS responses in the first three phases of treatment. In contrast, the OLS scores of the patients were significantly higher in Phases 1 and 2, coinciding only in the third phase. Finally, in the fourth phase of therapy, both the POLS and the OLS of the patients were well below the therapist’s estimate, both being slightly lower than those provided by the patients in Phase 3.

In terms of statistical significance, differences between any of the measures in Stages 1 to 3 were not significant. However, in Stage 4, the discrepancies between therapist and patients were significant (*p* = 0.038 for OLS and *p* = 0.042 for POLS).

### 3.2. Objective 2

To test whether the therapist’s perception, the patients’ perceptions, and the instrument used to analyze patient satisfaction agreed or disagreed, the following statistical tests were carried out.

#### 3.2.1. Result 1

In a first analysis, two-by-two comparisons were made between the three variables as it is shown in Table 3, Table 4 and Table 5.

No significant differences were observed between the assessment provided by the patients and that provided by the therapist.

No significant differences were observed between the assessment provided by the therapist and that provided by the patients using POLS.

Significant differences were observed. Patient perceptions reported by OLS were higher than those reported by POLS.

#### 3.2.2. Result 2

The following Table 6 shows the correlations between the three variables.

The results showed a high correlation between patients’ assessments using the two measures: OLS and POLS. In fact, the coefficient of determination indicated that there was 57% shared variance (computed as the square of r).

In addition, a significant correlation (0.743) was observed between the therapist’s perception (OLS-Therapist) and that expressed by the patient through POLS, with a shared variance of 55%. To a lesser extent, the correlation between therapist satisfaction and the OLS expressed by patients was also significant. In this case, the shared variance was only 25%; that is, half the value observed between the therapist’s criteria and POLS.

## 4. Discussion

The first aim of the present article concerned the validation of the graphic scale by means of both an independent measure, provided by OLS-therapist (external validity), and a concurrent measure, provided by the OLS-patient measurement.

The results showed that the therapist’s assessment coincided with the responses provided by the patients for POLS in the first three phases of treatment. The responses provided by the patients in the form of OLS-patient coincided only in the third phase, being significantly higher in Phases 1 and 2. It should be noted that the greatest differences were observed in the fourth phase, where both the POLS and the OLS patient scores were significantly lower than the assessment made by the therapist. Moreover, both were slightly lower than those provided by the patients in Phase 3.

These results show that, once they reached a certain level or state, the patients perceived that they had reached a maximum level of satisfaction. These were subjective perceptions, as the data did not correspond to the scores provided by the therapist; therefore, the patients were able to advance to the last phase or level of treatment.

It is reasonable to consider the agreement between patient and therapist ratings in the first three phases as strong evidence for the accuracy of the instrument, although limited by the moderate sample. On the other hand, the discrepancies in Phase 4 were probably not due to the instrument itself, but rather indicative of the fact that the assessment of improvement by the patients did not take into consideration parameters that are important for completing the therapeutic process from a professional perspective. Furthermore, in normalizing their life situations, they had to face a set of circumstances that went beyond their main objectives in the previous months—physical recovery, changes in eating behavior, and psychological rehabilitation. In other words, until Phase 3 of treatment, the patients’ lives were very much focused on the therapy and the physical and psychological improvements resulting from it. In Phase 4, on the other hand, their lives shifted to other centers of interest (and potential problems), introducing more sources of dissatisfaction; importantly, these elements were outside the scope of the assessment conducted by the therapist, who remained focused on the problematic eating behavior. Consequently, even if improvements in the symptomatology of the treated disorder were consistent—and were reflected in the therapeutic evaluation—they were partially neutralized or reduced by the appearance of other, perfectly normal, sources of dissatisfaction. This agreed with the findings of other studies [40,41], which indicated that clinical ED populations experience lower life satisfaction than those in the general population.

With regard to concurrent validation, the comparison of the POLS and OLS scores for each patient revealed their high congruence, sharing 57% of the variance observed; however, the POLS values were almost indiscernible from the scores provided externally by the therapist, at least in the first three phases, while the OLS scores of the patients tended to be somewhat more optimistic during the first two phases, only matching the POLS scores and the therapist’s assessments in the third phase. Thus, the life course perspective and reflection associated with the completion of the graph seemed to favor greater accuracy in assessing the current state of satisfaction. On the other hand, when this assessment was carried out in an ad hoc manner, dissociating it from the life course perspective, it seemed to be more sensitive to elements of social conformity or acquiescence, tending towards a certain overestimation of wellbeing.

Overall, the results obtained pointed to the validity of the LSCh procedure. Of course, the verification was carried out only with the most recent values, since it was not possible to obtain reasonably solid elements of external validity for the past values. However, the high consistency with the therapist’s assessments, as well as the consistency with the patients’ own point assessments, detached from the life-course perspective, could be considered as evidence of validity (though the limitations imposed by the moderate sample size should be kept in mind).

In relation to the second objective—which concerned the greater accuracy of POLS compared to OLS—the results obtained indicated that the scores provided by the patients for OLS were significantly higher than those provided for POLS, the mean value of which coincided almost completely with the scores provided by the therapist.

Thus, there is reason to consider that the LSCh has considerable utility and validity for the diagnosis and treatment of patients suffering from EDs, just as it has proven its usefulness and potential in studies on wellbeing and life satisfaction [25,34,35] and life history research in the field of social sciences [24].

The graphic format and the perspective of the whole life course lead to increased objectivity; on the other hand, the isolated assessment is sensitive to many circumstantial factors, linked to the immediate temporal space of the moment of response, and is more prone to biases of acquiescence. Thus, if a very positive or negative event has occurred close to the moment of evaluation, it can distort the isolated information provided; conversely, this phenomenon rarely occurs when the evaluation is retrospective and is framed within a more global perspective, as this activates an orientation towards the average, in the sense of filtering out events that occur by chance and have a short-lived impact. As an example, it is quite normal when there has been a recent positively or negatively charged event to think that life is idyllic or a real disaster. However, when this same event is analyzed and compared with other events that have occurred at other times in life, it is very likely that the initial intensity will be lessened, regardless of the valency (positive or negative).

Furthermore, the high correlations found between the answers provided by the patients (verbally or through the graph) and those provided for POLS and the assessments made by the therapist highlight the congruence of the scale, at least as far as the POLS item is concerned. This validity cannot be automatically generalized to the assessments of previous moments in the life trajectory, since there are variables linked to memory and reconstruction that have almost no impact on the assessment of the present moment. However, it does indicate that, if the assessment of the present moment is suitably accurate, the graph as a whole works as an objective point of reference. Although the person may have some bias when assessing certain previous life stages, due to the distortions linked to memory, the comparison of these stages with the present moment makes it possible to correct, at least in part, these biases. It is important to note that the exercise involves not only drawing the graph, but also explaining arguments or reasons that justify the levels described. In this sense, the reflection included in the process serves to contextualize the assessment of each point in the life trajectory and allows for considerable adjustments to the description, for example, by giving weight to the designation of periods intuited as good or bad, but for which there are no obvious arguments to justify why they were considered as such.

The data collected by the present project were consistent with the findings of other studies on the subject. Firstly, as stated by [28,29], the results of this work underline the importance of including elements of visual evaluation that make it possible to go beyond an autobiographical account of a particular experience, of an interpretative nature, which does not support the verbal reconstruction of past events. This is an important advantage, especially in studies involving young participants, since Conijn et al. [42] observed that young people do not differentiate concepts that researchers and adults in general would consider distinct.

Furthermore, and in line with the arguments of [30], the LSCh as a means of graphic expression is a fundamental tool through which one may express what cannot be easily said through words, allowing oneself to be carried away by what one feels. By creating a drawing or graph, one can capture what one feels or needs to express, so that drawing or graphing becomes a tool to express emotions, experiences, and feelings. In social interaction, words are more loaded in terms of correctness and are more linked to moral evaluations. In contrast, a graph is a rather neutral starting point in which one’s own perception of wellbeing can be expressed without the use of words. True, it is necessary to verbally justify the reasons for the different evaluations, but the drawn graph is already a tangible element that has captured the initial situation, making it more difficult to omit explanations. On the other hand, with exclusively verbal protocols, it is easy to omit stages that are perceived as incongruous with socially dominant values or that may be considered inappropriate or pathological behavior.

Moreover, it should not be forgotten that visual formats prevail over verbal information, so it is likely that the level of motivation and involvement in this type of technique is higher than for answering a questionnaire. This motivation could also be used to meet therapeutic challenges and targets, along the lines of the arguments in [30,43].

### Limitations

The main limitation of this study was the nature of the sample, which was moderate in size and originated from a single therapeutic center and may therefore have contained socio-geographical or socio-cultural biases. However, the sample included the whole population of patients with eating disorders who were treated in the health center, so no filtering was applied by the researchers.

Another limitation was the lack of other objective tests with which to compare the ratings for past life stages obtained through the LSCh. This would require a longitudinal approach whereby scores provided for the present are contrasted with the assessment of the same timepoint after a certain period.

Notwithstanding, it should be noted that the value of the test does not lie so much in the objectivity of the assessments of previous moments, but rather in the way in which these stages are perceived by the person and the possibility of reflecting on them. In fact, the development of the life trajectory itself is a correction factor for these evaluations, so that the evaluation of the degree of wellbeing at a certain point depends on the other moments with which it is compared. However, in psychological terms, the way in which the respondent assesses their wellbeing, i.e., his or her cognition, often has a greater impact than the objective facts. Therefore, it is not an exercise of remembering and objectively assessing past life stages, but of elucidating one’s appreciation of these stages and weighing them up in an argumentative way.

## 5. Conclusions

The contributions of this work can be summarized as follows:(1)The availability of a graphical technique such as the LSCh, which reveals how a patient perceives “their subjective world” and “how they feel about their levels of wellbeing and discomfort” over time and according to past events/experiences, is of great interest as a therapeutic tool, both for understanding patients’ previous histories and for monitoring during therapy.(2)The POLS indicator was shown in this pilot study to have external and concurrent validity in therapeutic contexts, at least in the treatment of eating disorders. Confirming this result with a wider sample is advisable, although theoretical arguments concerning the contextualization of the response in the overall developmental trajectory suggest that this result was congruent and predictable.(3)The results indicated that, in this pilot study, the POLS score was more accurate and reliable than the decontextualized measures of life satisfaction (OLS-Patient) used in therapeutic contexts, at least in the treatment of eating disorders. This certainly must be confirmed in a wider and more varied sample, though the current results are promising.

Future research is needed to test the usefulness and validity of the LSCh and POLS techniques in further therapeutic contexts involving the treatment of other mental pathologies, as well as in contexts of mental health promotion and personal growth.

## Figures and Tables

**Figure 1 ijerph-19-14452-f001:**
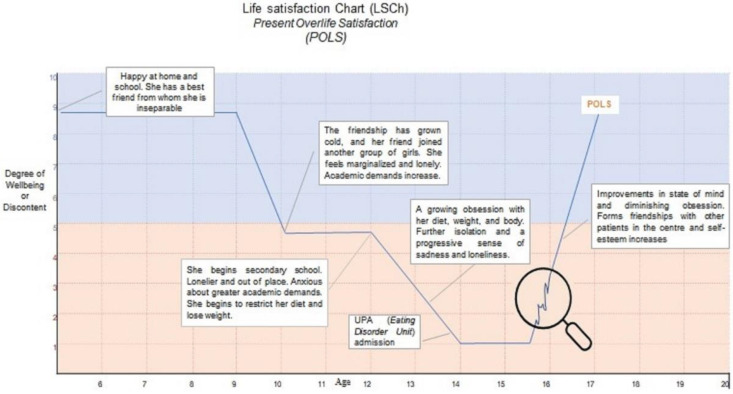
LSCh and POLS of a patient.

**Figure 2 ijerph-19-14452-f002:**
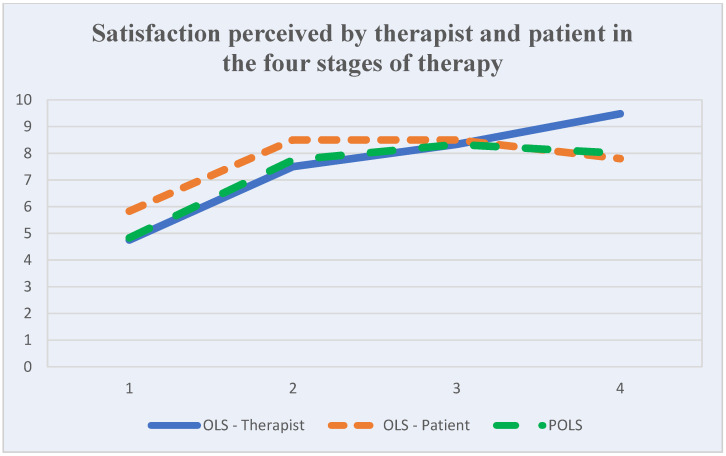
Satisfaction perceived by therapist and patients in the four stages of therapy.

**Table 1 ijerph-19-14452-t001:** Distribution of patients by level of disease progression.

Level	N (%)
1	12 (41.38)
2	6 (20.69)
3	6 (20.69)
4	5 (17.24)

**Table 2 ijerph-19-14452-t002:** Means and deviations by treatment level of the total population (* *p* < 0.05; ** *p* < 0.001).

	Level				
	1				
	N = 12	N = 6	N = 6	N = 5	Chi-Square
OLS-Therapist	4.75 ± 2.26	7.5 ± 0.55	8.33 ± 1.21	9.49 ± 0.89	18.65 **
OLS-Patient	5.83 ± 1.99	8.50 ± 1.05	8.50 ± 0.84	7.80 ± 1.09	12.67 *
POLS-Patient	4.83 ± 2.41	7.75 ± 0.76	8.33 ± 1.03	8.00 ± 1.50	14.67 *

**Table 3 ijerph-19-14452-t003:** N, mean, and standard deviation of OLS scores.

	N	Mean	Deviation	Z
OLS-Patient	29	7.28	1.91	−0.520
OLS-Therapist	29	6.86	2.46	

**Table 4 ijerph-19-14452-t004:** N, mean, and standard deviation of POLS-Patient and OLS-Therapist scores.

	N	Mean	Deviation	Z
POLS-Patient	29	6.71	2.35	−0.749
OLS-Therapist	29	6.86	2.46	

**Table 5 ijerph-19-14452-t005:** N, mean, and standard deviation of the POLS-Patient and OLS-Patient scores (* *p* < 0.05).

	N	Mean	Deviation	Z
POLS-Patient	29	6.71	2.35	−2.434 *
OLS-Patient	29	7.28	1.91	

**Table 6 ijerph-19-14452-t006:** Correlations (** *p* < 0.001).

	POLS-Patient	OLS-Patient
OLS-Patient	0.757 **	
OLS-Therapist	0.743 **	0.504 **

## Data Availability

The data presented in this study are available on request from the corresponding author.

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
