# Peer review of "Efficacy of a Contextualized Measurement of Life Satisfaction: A Pilot Study on the Assessment of Progress in Eating Disorder Therapy"

_ijerph, 2022, doi:10.3390/ijerph192114452_

Round 1
Reviewer 1 Report
Thank you very much for the opportunity to review this interesting manuscript.
below my comments:
- In my opinion, the use of the concepts of quality of life and satifaction of life interchangeably is wrong. Because life satisfaction is part of your quality of life. Please explain.
- please provide information on the research that the authors refer to (e.g. on which group they were carried out, number of people)
- please complete the number of therapists included in the study
- please explain why the rPearson correlation was used, if the entire study was measured with non-parametric tests
- references contains little of the latest research reports
- a small number of respondents were included in the limitations, however, the statistical methods used do not allow for further reaching conclusions. in my opinion, this study would benefit if only a qualitative description of individual cases was used.
Author Response
Dear reviewer,
We are grateful for all the comments and suggestions provided. We are sure that they have contributed to a significant improvement in our work.

Reviewer 2 Report
Dear Authors,
A review of the literature on quality of life research reveals many inconsistencies, both in defining and understanding the term, which covers a wide range of human existence. The term "quality of life" is such a scientific construct that it is difficult to define unambiguously and establish a common position on how it is perceived and understood due to its high generality. Any neglect of the definition leads to inaccuracies and terminological misunderstanding. Assessment of one's own health from the patient's point of view, is considered an important element in the therapeutic process. Patients usually perceive illness as a condition caused by external factors, forgetting that many of the relevant risk factors are related to their lifestyle and mindset. The interconnection between the physical and psychological zones proves that a person's thoughts and behaviors significantly affect his or her well-being, and thus the broader quality of life.
The manuscript presented to me for evaluation despite, an initial good impression, quickly revealed many methodological inaccuracies.
Do not use abbreviations in the executive summary when they are not developed until the introduction.
In addition to the small sample size, which is undoubtedly a downside, the study lacks inclusion criteria. The sample selection was purposive, being patients from a single center. Women with diagnosed eating disorders What kind of eating disorder? At least in this context, was the group homogeneous, or do we have a division into several smaller groups and different diagnoses, no indication of the diagnosis (diagnostic criteria), which makes it even more difficult to reliably draw conclusions from the study.
The statistical sample starts with 30 individuals, so it is too small to generalize conclusions. There is also a lack of reliable, broad characterization of the study group, which the manuscript boils down to the average age of the patients. There is no information about the local Bioethics Committee's approval of the study.
Please demonstrate the psychometric values of the scale. No comparison of the results with another scale, so how to prove that it is better (more sensitive).
Section 3.1 Statistical Analysis Overview should be found in the Material and Method section.
The conclusions are too generalized. It might be worthwhile to treat the study as a pilot and reflect this appropriately in the title and method of the study.
The analysis and evaluation of quality of life, despite the general criteria adopted, as well as detailed guidelines to facilitate the selection of appropriate measurement tools, is not simple, as there is no universal rule in this area. Tools found to be useful in a selected clinical situation may prove to be inappropriate and undesirable in other studies. The quality of life questionnaires that constitute measurement scales were created to obtain variables that pose research difficulties, however, the use of the tools presented does not convince me to accept the current form of the manuscript and needs to be improved.
Greetings
Author Response
Dear Reviewer,
We are grateful for all the comments and suggestions provided. We are sure that they have contributed to a significant improvement in our work.

Round 2
Reviewer 1 Report
Thank you very much to the authors for taking into account my suggestions
Reviewer 2 Report
Dear Authors,
The changes made by the authors significantly improve the quality of the manuscript.
Greetings!